# A Current Update on the Role of HDL-Based Nanomedicine in Targeting Macrophages in Cardiovascular Disease

**DOI:** 10.3390/pharmaceutics15051504

**Published:** 2023-05-15

**Authors:** Alankrita Rani, Gunther Marsche

**Affiliations:** 1Division of Pharmacology, Otto Loewi Research Center, Medical University of Graz, Neue Stiftingtalstrasse 6, 8010 Graz, Austria; alankrita.rani@medunigraz.at; 2BioTechMed-Graz, Mozartgasse 12/II, 8010 Graz, Austria

**Keywords:** lipoprotein, apolipoprotein, HDL, nanoparticle, nanodisc, inflammation, monocyte, macrophage, atherosclerosis, cardiovascular

## Abstract

High-density lipoproteins (HDL) are complex endogenous nanoparticles involved in important functions such as reverse cholesterol transport and immunomodulatory activities, ensuring metabolic homeostasis and vascular health. The ability of HDL to interact with a plethora of immune cells and structural cells places it in the center of numerous disease pathophysiologies. However, inflammatory dysregulation can lead to pathogenic remodeling and post-translational modification of HDL, rendering HDL dysfunctional or even pro-inflammatory. Monocytes and macrophages play a critical role in mediating vascular inflammation, such as in coronary artery disease (CAD). The fact that HDL nanoparticles have potent anti-inflammatory effects on mononuclear phagocytes has opened new avenues for the development of nanotherapeutics to restore vascular integrity. HDL infusion therapies are being developed to improve the physiological functions of HDL and to quantitatively restore or increase the native HDL pool. The components and design of HDL-based nanoparticles have evolved significantly since their initial introduction with highly anticipated results in an ongoing phase III clinical trial in subjects with acute coronary syndrome. The understanding of mechanisms involved in HDL-based synthetic nanotherapeutics is critical to their design, therapeutic potential and effectiveness. In this review, we provide a current update on HDL-ApoA-I mimetic nanotherapeutics, highlighting the scope of treating vascular diseases by targeting monocytes and macrophages.

## 1. Introduction

High-density lipoproteins (HDL) are a naturally occurring class of nanoparticles comprising a dynamic conglomerate of lipids, proteins and other biomolecules [1]. The correlation between HDL and incidence of disease dates back to the 1950s when lipoprotein cholesterol content was first identified as a risk factor for cardiovascular disease, which is followed by the elaborate Framingham study in the 1960s [2,3,4,5]. An inverse relation has been established between HDL-C content and CAD. Since then, there has been a considerable advancement in the field of HDL-based theragnostic strategies in cardiovascular disease amelioration with about 1300 clinical trials registered so far that directly or indirectly harness HDL metabolism and function (clinicaltrials.gov (accessed on 29 March 2023); search keywords: cardiovascular diseases and HDL). However, HDL cholesterol raising therapies did not unequivocally succeed in modulating disease progression to the extent predicted [6,7].

The lack of undebated evidence of far-reaching benefits [8,9] from HDL cholesterol-raising therapies has led to the need for a better understanding of HDL structure, metabolism and functional attributes. Native HDL has multiple protective effects in vascular disease owing primarily to cholesterol efflux, anti-thrombotic, anti-oxidant and anti-inflammatory effects on monocytes, macrophages and endothelial cells [10,11,12]. Numerous studies support that enhancing macrophage cholesterol efflux by HDL particles, the first step of reverse cholesterol transport (RCT), promotes the regression of atherosclerotic plaque and restores vascular health. HDL remodeling in disease, loss of function and shift to pro-inflammatory type have been implicated in several diseases [13,14,15,16,17,18,19,20,21,22,23,24,25,26,27,28,29,30], reiterating the need to refurbish HDL particle quality and quantity.

Advances in proteomics and lipidomics, as well as structural analyses and a variety of approaches to determine the functional “quality” of HDL particles, have deepened our understanding of the factors that render HDL dysfunctional or even pro-inflammatory in certain diseases [31,32]. It is now generally accepted that inflammation leads to a loss of the anti-inflammatory characteristics of HDL. During inflammation, HDL-associated apolipoprotein A-I (ApoA-I) is replaced by acute phase proteins such as serum amyloid A (SAA) and rendered dysfunctional by myeloperoxidase-derived reactive oxidation species [17]. This has led to the development of second-generation HDL mimetics, with the majority of HDL nanoparticles being engineered with ApoA-I as the scaffold [33,34,35,36]. ApoA-I, the major protein component of HDL, is largely responsible for regulating reverse cholesterol trafficking and protecting against cardiovascular disease, and it is also known to modulate inflammatory and immune responses. Such synthetically designed HDL nanoparticles have shown immense pre-clinical potential and have now advanced to various phases of clinical trials [37,38,39]. Apart from the applications in inflammatory settings of vascular disease, current advancement in HDL-based nanomedicine is also evident in the fields of autoimmunity and immuno-oncology due to their biocompatibility, functional pleiotropy and structural feasibility for optimal drug design [36].

In this review, we provide an up-to-date overview of the latest ApoA-I-based HDL nanomedicine approaches for the treatment of cardiovascular disease, targeting the mononuclear phagocyte system.

## 2. The Mononuclear Phagocytes Orchestrate Vascular Inflammation and Remodeling

Monocytes are mononuclear phagocytic cells of the myeloid lineage, which play indispensable roles in host innate immunity and homeostasis [40,41]. Originating from the bone marrow and primarily circulating in the blood, monocytes play critical roles in surveillance and response to pathogens, injury and inflammation. Owing to their plastic and diverse transcriptional profile, which allows them to take on different roles under homeostatic and pathogenic conditions, monocytes differentiate to phenotypically distinct and specialized macrophage subclasses. Apart from the inherent functions performed by tissue resident macrophages, the migration and expansion of monocytes to the inflamed tissues eventually decides the course of disease progression and its containment [42]. Based on the cytokine and chemokine profile of their extracellular milieu, monocytes and monocyte-derived macrophages orchestrate repair, remodeling, inflammation and its resolution in vascular diseases, thereby becoming excellent drug targets [43,44,45,46].

Coronary artery disease is a cardiovascular disorder that progresses with the unique, detrimental interplay of metabolism and immunity [47,48]. Continuous exposure to associated risk factors such as unhealthy diet, high blood pressure, diabetes, smoking, high low-density lipoprotein cholesterol, physical inactivity, etc. ultimately leads to a chronic inflammatory state that compromises vascular health. Atherosclerotic tendency and metabolic syndrome are being increasingly reported as interrelated complications [49]. In cardiovascular disease, lipid deposition and associated endothelial cell damages such as increased Vascular Cell Adhesion Molecule-1 (VCAM-1) and Monocyte Chemoattractant Protein-1 (MCP-1) expression are significant contributors that set the stage for further aggravation. In response to this, patrolling monocytes are recruited to the intimal layers of the endothelium as they differentiate to lipid-engulfing foam cells, leading to the formation of atherosclerotic plaques [50]. The tissue resident macrophages also respond to hyperlipidemia by differentiating into foam cells. Apart from foam cell formation, polarization is a critical step that further regulates disease severity by influencing the cytokine profile in the plaque. The non-foam macrophages contribute greatly to the inflammatory cytokine production in the plaque [51]. Several macrophage subclasses have been reported in the atherosclerotic plaques, the majority of which are of pro-inflammatory and pro-atherogenic nature. The classical M1 macrophages constitute about 40%, M2 macrophages constitute about 20%, Mox macrophages constitute about 30% and the remaining 10% macrophages have a distinct cytokine profile and protective features [52]. M1 macrophages sustain inflammation, and their apoptosis triggers further immune cell infiltration and neutrophil and platelet activation. The ratio of M2:M1 macrophages is directly proportional to plaque stability due to the protective activities and pro-resolving functions of M2 macrophages [52]. Mox macrophages are polarized by oxidized phospholipids and express redox-regulatory genes and Toll-like receptor-2 (TLR2)-mediated inflammatory cytokines. Mox macrophages promote M1 polarization and are highly proatherogenic [53]. Single-cell RNA sequencing in mice and human atherosclerotic plaques has added new insights to the traditional classification of macrophage subtypes. Some of the studies have also identified new subclasses of macrophages and improved our understanding of their origin. Some newly discovered subtypes include CD163-positive, hypoxia-inducible factor -1α and vascular endothelial growth factor positive macrophages or smooth muscle derived, proliferative, interferon-inducible and cavity macrophages [54,55]. Ongoing inflammatory insults [56], extracellular matrix deposition [57] and altered hemodynamics can trigger thrombus formation and rupture of the plaque, leading to myocardial infarction and ischemic stroke. The central role of monocytes in atherosclerosis, their recruitment and differentiation to foam cells has been greatly emphasized and targeted for the prevention of atherosclerosis [56,58,59,60].

## 3. High-Density Lipoproteins Are Self-Assembling Nanoparticles

Plasma lipoproteins are complex macromolecules composed of lipids and proteins which are classified into subclasses based on their size (5 nm up to 1200 nm) and density (0.930–1.20 g/mL) [1]. Of the several classes of lipoproteins [61], HDL are of utmost importance for lipid homeostasis and additionally have multiple protective effects [62]. HDL are naturally classified as nanoparticles in circulation, given their nanoscale size of 5–12 nm, with a density ranging from 1.063 to 1.21 g/mL. HDL subclass distribution and classification are dependent on the isolation and separation methods [63]. After isolation by ultracentrifugation, two HDL subclasses can be distinguished on the basis of their density: HDL3 (1.125–1.21 g/mL) and HDL2 (1.063–1.125 g/mL) [1]. Using NMR, HDL is classified into three and up to five subtypes based on particle diameter: large HDL (8.8–13.0 nm), medium HDL (8.2–8.8 nm) and small HDL (7.3–8.2 nm) [63,64,65,66]. Lipid poor forms of HDL are small and discoidal, whereas a fully mature HDL particle is spherical, larger and carries additional hydrophobic cargo. The method used for HDL isolation substantially impacts the functional readouts, as each method has its advantage and specific enrichment of some proteins when compared to the others [67]. Primarily involved in the reverse cholesterol transport from peripheral tissues to the liver, HDL particles have additional anti-oxidant, anti-inflammatory, anti-thrombotic, and anti-apoptotic properties. The pleotropic immunomodulatory effects and diverse functionality of HDL is well documented and extensively reviewed [12,68]. The diverse and complex composition of HDL ensures its endogenous characteristics as a natural, selective nanotherapeutic entity.

### 3.1. HDL Composition and Function

Since its first isolation in 1929 [69], properties and functions of HDL have been thoroughly investigated [70]. HDL particles are able to influence a plethora of biological functions because of their multi-molecular and diverse structural arrangement. They are composed of a variety of proteins, enzymes and lipids. Apart from this classical molecular complexity, the existence of different microRNAs in association with HDL expands the magnitude of their biological influence to an even higher degree [62,71]. An overview of HDL components highlighting the expanse of HDL function in cardiovascular disease is summarized in Figure 1.

#### 3.1.1. Protein Components

##### Apolipoproteins

The primary sources of HDL-associated apolipoproteins are the liver and intestine. Their major functions include providing structural feasibility for the assembly of HDL nanoparticles in vivo, interaction with specific lipoprotein receptors that imparts tissue-specificity and as activators or inhibitors in enzymatic reactions [1,63]. HDL particles are highly enriched in apolipoprotein A-I, accounting for approximately 70% of the total HDL protein. Other major apolipoproteins include ApoA-II, ApoA-IV, ApoCs, ApoD, ApoE, ApoF, ApoH, ApoJ ApoL-I, ApoM and ApoO, each with a unique function discovered and documented in numerous proteomic studies [72]. A recent review compares 37 different investigations of HDL proteome summarizing around 566 proteins that are found in association with HDL [73].

The self-assembly of HDL as definitive nanoparticles is largely attributed to the structural uniqueness of ApoA-I. HDL biogenesis begins with the secretion of ApoA-I from the liver and intestine into the bloodstream, where it binds phospholipids forming pre-beta discoidal HDL particles. Being composed of several amphipathic helices, ApoA-I is able to simultaneously interact with the phospholipid acyl chain and the aqueous environment. This initiates and maintains the initial nano-discoidal and later on mature nano-spherical structure of HDL. The three-dimensional structure of ApoA-I bound to phospholipids has been thoroughly explored [74]. The structural properties of ApoA-I are harnessed in the synthesis of HDL nanotherapeutics as a naturally available blue print for design [75,76]. Apart from offering structural advantages to HDL nanoparticles as a scaffold for stability, HDL-associated ApoA-I is responsible for a majority of anti-inflammatory and atheroprotective properties of HDL, especially in promoting cholesterol efflux and preventing the migration of monocytes to the inflamed endothelia [77,78,79,80,81]. ApoA-I gene transfer in mice reduces TLR4 expression on the endothelial cells, which further regulates the expression of ICAM-1 and VCAM-1, hence controlling leucocyte migration across endothelia in response to inflammation [82]. The positive characteristics of ApoA-I, which underline its potential as a therapeutic candidate, have been described [83,83]

ApoE is another class of apolipoproteins whose functions are critical for lipid homeostasis [84]. Apart from liver, macrophages in the arterial wall also produce ApoE. Distributed across lipoprotein classes, ApoE interacts with low-density lipoprotein receptor (LDLR) and enables the clearance of triglyceride-rich lipoproteins and HDL. It shares structural similarities with ApoA-I in terms of amphipathicity and interacts with ABCA-1 to generate nascent HDL particles. HDL particles associated with ApoE can be larger than those associated with ApoA-I, which is due to the differences in their interaction with the phospholipids, making it more capable of adjusting to the core expansion of the HDL upon cholesterol esterification [85]. The ApoE-mediated mobilization of cholesterol and catabolism of triglycerides is employed in ApoE^-/-^ mice, which rapidly develop CAD induced by dyslipidemia when fed a high-fat diet [86]. The impact of ApoE on HDL in clinical contexts remains unclear, perhaps because its true effect may be obscured by other proteins. Nevertheless, among five proteins that differ between individuals with coronary artery disease and control subjects, ApoE was found to be the most prevalent protein in HDL obtained from atherosclerotic lesions, indicating its potential significance in the pathophysiology of atherosclerosis. However, studies examining the association between ApoE in HDL and cardiovascular disease outcomes have been inconclusive [87,88].

ApoC-II, a cofactor for lipoprotein lipase (LPL) activation, participates in lipid homeostasis by enabling triglyceride hydrolysis [89]. Apolipoprotein C-II contains three amphipathic helices. In normolipidemic subjects, ApoC-II was found to be mainly distributed in HDL, whereas in hyper-triglyceridemic subjects, it is predominately distributed in VLDL and LDL particles [90]. As ApoC-II levels increase, the size of HDL particles decreases, suggesting RCT is reduced and HDL maturation is hindered. ApoC-II alone may be a factor that influences the distribution of HDL subclasses. In addition, the size of HDL particles was found to correlate with the ratio of ApoC-III to ApoC-II [91]. Monogenic disorders in the *APOC2* gene are related to extreme elevations in triglycerides and chylomicronemia [92]. These features have sparked a renewed interest in using ApoC-II as a drug candidate.

ApoC-III is an inhibitor of LPL produced mainly by the liver and intestine, and approximately half of plasma ApoC-III associates with HDL. Elevated levels of ApoC-III in the blood predict the risk of cardiovascular disease in humans [93], whereas ApoC-III loss variants in humans are associated with a cardioprotective effect [94]. The mechanisms by which ApoC-III increases the risk of atherosclerotic cardiovascular disease are the subject of intensive research. Excessive incorporation of ApoC-III can render HDL dysfunctional by impairing HDL-mediated cholesterol efflux capacity and promoting inflammation and organ damage through alternative inflammasome activation [95]. Recent data suggest that the atherogenic effects of ApoC-III are principally mediated by slowing the clearance of triglyceride-rich lipoproteins rather than by increased inflammasome activation [96]. The inhibition of ApoC-III is a therapeutic approach under consideration for the treatment and prevention of cardiovascular diseases [97]. Approximately 13% of the total HDL particles contain ApoC-III [98]. There are opposite associations between HDL-C levels with and without ApoC-III and the risk of CHD in studies that follow apparently healthy men and women over time [98].

The interaction between ApoE and ApoC-III on HDL has implications for both metabolism and coronary heart disease [99]. A recent study showed that ApoE promotes metabolic processes involved in reverse cholesterol transport, which is associated with a lower risk of coronary heart disease. However, the presence of ApoC-III on HDL with ApoE abolishes these benefits. Consequently, differences in the metabolism of HDL subtypes, particularly in terms of reverse cholesterol transport, are reflected by differences in their association with coronary heart disease [99].

##### HDL Associated Enzymes

HDL carries many enzymes which primarily impart specific biological functions and enable their optimal structure configuration [1]. One of the important enzymes include lecithin-cholesterol acyltransferase (LCAT), which is particularly important in promoting the maturation of HDL particles. The continuous esterification of free cholesterol and its incorporation into the core of the HDL particles enables a more spherical morphology of the nanoparticles and creates space for further cholesterol incorporation into the particles. ApoA-I interacts with LCAT and catalyzes the esterification step, enabling the cardioprotective functions of HDL.

HDL-associated esterases mostly belong to the paraoxonases family (mainly PON-1 and PON-3). These calcium-dependent enzymes reduce oxidative stress burden by preventing lipoprotein oxidation. In addition, PON-1 modulates monocyte transmigration through the endothelium [100]. It also enhances macrophage cholesterol efflux via ATP-binding cassette 1 (ABCA-1), and its deficiency leads to atherosclerotic complications [101,102]. In a recent report, the selective expression of ApoA-I and PON-1 has been shown to alleviate TNF-α induced endothelial dysfunction by reducing inflammatory and oxidative stress [103]. PON is recognized as a labile enzyme, which is quickly broken down and loses its function under specific circumstances. However, when PON is associated with HDL, it is protected from degradation and retains its enzymatic activity [104,105]. PON can also affect the stability of HDL. Research has shown that when PON-1 is associated with HDL, it can increase the structural stability of the lipoprotein particle. This in turn may protect it from oxidative destruction and prevent the development of atherosclerotic plaques. Together, the presence of ApoA-I, LCAT and PON-1 prolongs the time that HDL can inhibit LDL oxidation in part by preventing LCAT inactivation [106,107].

Another important enzyme, although more enriched in the LDL subclass, includes lipoprotein-associated phospholipase A_2_ also called platelet-activating factor acetyl hydrolase, which imparts anti-inflammatory and anti-thrombotic properties to HDL by specifically hydrolyzing the inflammatory mediator platelet activating factor and degrading lipid oxidation products [1,72].

In the plasma, HDL is also associated with an extracellular antioxidant enzyme glutathione peroxidase-3, which regulates the oxidative stress burden by reducing hydro-peroxidases. The role of such peroxidases is indispensable in the prevention of hypertension and cardiovascular complications and the maintenance of a biological redox state [108,109].

##### Enzymes and Proteins Partially Associated with HDL

Cholesteryl ester transfer protein (CETP) is responsible for a dynamic exchange of lipid species between the HDL, LDL and chylomicron subclasses. It enables the movement of cholesterol esters and triglyceride molecules between HDL and VLDL particles, enabling a constant interaction between the lipoproteins for lipid homeostasis. The genetic deficiency of CETP in human and animals showed a reduced atherosclerotic tendency with higher HDL-C levels, which led to the development of several clinical trials testing different CETP inhibitors; however, this demonstrated only limited benefits [110].

Phospholipid transfer protein exchanges phospholipids between HDL particles and other triglyceride-rich lipoproteins. It regulates the size of HDL particles in circulation by regulating cholesterol esterification by ensuring phospholipid supply. It is also known to interact with ApoA-I [111].

In case of an inflammatory response and in atherosclerosis, acute phase proteins such as serum amyloid a (particularly SAA-1), which is secreted by the liver, associates with HDL and modifies its function. Depending upon the concentration and the range of the acute phase response, SAA can displace ApoA-I from the HDL particles, thereby altering the structure and protective properties of HDL [112]. 

Other protein components of HDL that contribute to its immunomodulatory and homeostatic functions include the LPS-binding protein, complement proteins C3 and C4, and enzyme inhibitors such as α1-antitrypsin, α2-antiplasmin, retinol binding protein, platelet basic protein, vitronectin, etc. [70].

#### 3.1.2. HDL-Associated Lipids and Steroids

The HDL-associated lipids are mainly responsible for the hydrophobic and amphiphilic characteristics. Apart from the biophysical advantages of supporting particle structures and lipid homeostasis, the lipid components of HDL influence many anti-inflammatory properties of HDL.

Phospholipids are amongst the most abundant lipid groups in HDL, of which phosphatidylcholine is the most abundant, followed by lysophosphatidylcholine (LysoPC), phosphatidylethanolamine and plasmalogens. Phospholipids play a critical role in both acute and chronic inflammation by acting as substrates for the oxidation and/or synthesis of pro- and anti-inflammatory mediators [113,114]. The importance of LysoPC in the context of inflammation and allergy is being increasingly reported, which reiterates the possible role of HDL lipids in disease amelioration [115]. The interaction of LysoPC with granulocytes in physiological conditions enhances anti-inflammatory effects [116,117].

Sphingolipids, namely sphingomyelin, ceramide and sphingosine-1-phosphate confer numerous bioactive characteristics to HDL. Sphingosine-1-phosphate (S1P) is a bioactive sphingolipid specifically bound to ApoM in HDL. Its interaction with S1P receptors on leucocytes and endothelial cells is associated with its vascular protective properties [118]. ApoM-bound S1P is highly efficient in maintaining endothelial barrier integrity and reducing the surface expression of ICAM-1 and E-selectin, thereby regulating monocyte migration across inflamed endothelial cells [118]. Considerable emphasis has been placed on S1P-based therapies that enhance and rescue HDL functionality in cardiovascular and metabolic diseases [119].

Among steroids and neutral lipids, cholesterol, cholesteryl esters and triglycerides are predominantly present and regulate the size and maturation of the HDL particles [120].

#### 3.1.3. Other Constituents

The association of micro-RNAs with HDL particles has opened new dimensions for exploring the effect of HDL quality and quantity in disease pathophysiology. So far, miR-33a, miR-30c, miR-92a, miR-122, miR-125a, miR-126, miR-145, miR-146a, miR-150, miR-155, miR-223, miR-378, miR-486, miR-17/92 and mir181c-5p have been associated with HDL particles [121,122]. 

## 4. Anti-Inflammatory Effects of Native HDL: Focus on Mononuclear Phagocytes

Several mechanisms have been reported for the immunomodulatory and protective functions of HDL and ApoA-I. HDL interacts with structural cells as well as immune cells in circulation, influencing both innate and adaptive immunity [123]. The atheroprotective properties of HDL are primarily attributed, but not limited to, its interaction with endothelial cells, monocytes, macrophages, neutrophils and platelets. Cholesterol efflux and reverse cholesterol transport are the most discussed biochemical mechanisms that have inspired the development of HDL-based nanopharmaceuticals. In the exoplasmic side of the plasma membrane, specialized domains called lipid rafts function as a seat of several receptor-mediated signal transductions [124]. Consisting of cholesterol and sphingolipids, they are highly dynamic and regulate several cellular processes. ApoA-I/HDL-mediated cholesterol efflux profoundly affects the structure and activity of lipid rafts.

### 4.1. Anti-Inflammatory Mechanisms of HDL and ApoA-I

In a cardiovascular disease setting, the migration of monocytes to intimal layers of the endothelia is a major decisive point for disease progression. This transmigration is under the influence of chemotactic factors (e.g., MCP-1, M-CSF, etc.) from activated endothelial cells, which attracts the patrolling monocytes. In the cascade of events that lead to monocyte recruitment, the upregulation of monocyte integrins and adhesion factors are a critical requirement. One of the major integrins that facilitate monocyte trafficking across the endothelial layers is CD11b–CD18 complex. Upon treatment with isolated HDL or ApoA-I, CD11b expression is significantly reduced, suppressing monocyte adhesion to human coronary aortic endothelial cells and migration. The suggested mechanism highlighting the significance of cholesterol efflux mechanisms involves the collapse of lipid rafts by HDL or ApoA-I in an ABCA-1-dependent manner, leading to anti-inflammatory effects [125]. HDL and Apo A-I also reduced an MCSF-1-induced CD11 class of integrin expression, significantly inhibiting monocyte chemotaxis and spreading. Cell division control protein 42 is identified as a downstream target which links the efflux of cholesterol to cytoskeletal rearrangements in the monocyte [126].

Diabetes is one of the common comorbidities associated with CAD. The overexpression of ApoA-I in a diabetic mouse model suppressed hyperglycemia-driven myelopoiesis. ApoA-I abundance led to a general reduction in recruited monocytes in the transplanted atherosclerotic aorta in diabetic mice [127]. Human plasma-derived HDL infusion also promoted the M2 phenotype in lesion-associated or bone marrow-derived macrophages [127,128].

In addition to promoting cholesterol efflux, HDL induces activating transcription factor 3 (ATF3), which is a master regulator of immune response and homeostasis. Increased ATF3 binding across the chromosome suppresses inflammatory cytokine production and supports the epigenetic control of inflammation by HDL. ATF3 was shown to be critical for anti-inflammatory effects of HDL [129].

Type I interferon gamma response genes regulated by Toll-like receptor-4 (TLR4) were shown to be regulated by HDL. HDL inhibits the TIR-domain-containing adapter-inducing interferon-β (TRAM/TRIF) branch of TLR4 signaling by inducing TRAM localization to the intracellular compartment of murine macrophages, reducing the production of key inflammatory cytokines [130]. 

CD40/CD40 L is implicated in cardiovascular disease for the production of pro-inflammatory cytokines via NF-κB translocation [131]. In vitro studies on THP-1 macrophages showed that ApoA-I abolishes the CD40–TNF receptor-associated factor 6 interaction by modulating lipid raft structure and composition through ABCA1 mediated cholesterol efflux [132]. 

Dysregulation of the janus kinase/signal transducer and activator of transcription (JAK/STAT) signaling pathway is associated with various cardiovascular complications [133]. The JAK2–STAT3 pathway operates in close connection with ABCA-1, modulating its interaction with ApoA-I [134]. ApoA-I binding to ABCA-1 induced STAT3 phosphorylation and suppresses inflammatory cytokine (IL-6, IL-1β, etc.) production in mouse macrophages [135]. This finding, supported by other studies [134,136], establishes a clear link between cholesterol efflux and inflammation, placing ABCA-1 in the center of the mechanisms involving HDL and ApoA-I therapies that target vascular inflammation and myeloid cell malfunction [137,138].

### 4.2. Other Anti-Inflammatory Components of HDL

HDL-associated paraoxonases have been associated with numerous anti-inflammatory effects on mouse macrophages [139]. PON-1 transgenic mice show enhance protection against oxidative stress under hyperglycemic conditions [140]. The treatment of PON-1 knockout mice-derived macrophages with isolated human PON-1 significantly reduced the superoxide anion release and capacity to oxidize LDL [141]. PON-1 treatment also decreases cholesterol biosynthesis in murine macrophages [142]. PON-3 is associated with improved macrophage cholesterol efflux capacity in atherosclerotic mice [143]. 

S1P content of HDL has been shown to influence its cardioprotective activities in mice. In an ischemia reperfusion–injury model, S1P and HDL administration reduced leukocyte adhesion to the endothelium and cardiomyocyte apoptosis. The effects were nitric oxide dependent and mediated by the S1P3 receptor [144]. This also points toward the significance of the HDL nanodisc-lipid component when engineering optimally functional nanoparticles.

## 5. HDL Particles Are Remodeled in Cardiovascular Disease and Lose Their Function

There exists a strong structure–function relationship of HDL in several diseases. Both particle quality and quantity are greatly influenced by disease, environment and lifestyle. The past few decades have seen an overwhelming amount research reporting modification in HDL proteome, lipidome and biological functions in pathological conditions [13,14,15,16,17,18,19,20,21,22,23,24,25,26,27,28,29]. The alteration of HDL structure and function in disease is both an important biomarker of the diseased state as well as a potential critical factor influencing disease progression and regression [145,146]. Many post-translational modifications associated with inflammatory diseases render HDL-associated components dysfunctional and hence disrupt its normal functionality [147]. Figure 2 summarizes the modifications of selected HDL constituents in cardiovascular disease pathophysiology, highlighting its importance as a cause (therapeutic target) or consequence (diagnostic biomarker) of the disease.

Reductions in HDL-C levels are associated with an increased risk of CAD, and this principle formed the basis of first HDL-C-raising therapies [148]. Reduced levels of phospholipids and the total cholesterol content of HDL are consistently reported in inflammatory diseases [149]. Functional properties of HDL, such as the cholesterol efflux capacity, is altered in several inflammatory diseases and shows a strong correlation with subclinical and obstructive CAD independent of HDL-C levels [150,151]. It is thought that the abundance of smaller, denser HDL subfractions, which are considered more cardioprotective [152,153,154], are reduced in the atherosclerotic disease setting [155]. The level of HDL-triglyceride has a direct link to the metabolism and vascular changes that occur in arteriosclerosis. Higher HDL triglyceride levels can be viewed as an indicator of metabolic and cardiovascular risks [156].

Oxidative reactions and post-translational modifications, such as carbamylation, nitration, chlorination, etc. lead to a profound decline in HDL particle quality. The highly oxidative microenvironment in the atherosclerotic lesions is attributed, in part, to excessive immune cell infiltration and increased effector functions such as the release of extracellular traps and oxidative enzymes, e.g., myeloperoxidases, primarily from activated neutrophils [157,158]. The oxidative modification of HDL phospholipids leads to increased particle polarity and alters the anchoring of ApoA-I, which is manifested as lipid-depleted, cross-linked, dysfunctional ApoA-I found in atherosclerotic plaques [159]. It has also been hypothesized that HDL is a selective target for MPO-catalyzed oxidation and carbamylation due to the enrichment and co-localization of MPO-catalyzed oxidation signatures with malfunctional ApoA-I in the atherosclerotic plaques [160,161,162,163]. The oxidation-induced cross-linking of lipid poor ApoA-I in CAD lesions, reduced cholesterol efflux capacity, and the decline in PON-1 and LCAT activity of the HDL result from MPO enrichment in the atherosclerotic plaques [164,165]. Apart from oxidative modification, the replacement of ApoA-1 with SAA due to the chronic inflammatory conditions associated with CAD further reduces the protective features of HDL such as wound healing, proliferation and antioxidative effects [149].

The activation of endothelial nitric oxide synthase (eNOS) is one of the most important endothelial protective properties of HDL. HDL particles isolated from CAD and diabetes patients showed limited and even inhibitory activity for endothelial repair via the PKCβ-2 and Akt pathway [166,167,168]. This loss of function is thought be attributed to the inhibition of PON-1 activity in CAD, which might impair eNOS phosphorylation [166]. There is now strong evidence that HDL is rendered dysfunctional and sometimes even pro-inflammatory in CAD [149,169,170,171]. The loss of structure and function of HDL in CAD is at least partially rescuable with lifestyle changes and pharmacological interventions [172,173].

## 6. The Era of HDL Mimetic Nanoparticles: Replenishing HDL Quality and Quantity

With the shift in approach from simply raising HDL-C levels to improving HDL function, considerable emphasis has been put on restoring the structure and mechanistically augmenting atheroprotective properties of HDL [172]. The term ‘HDL mimetics’ refers to a class of isolated ApoA-I or mimetic peptides complexed with phospholipids or ApoA-I mimetic peptide-based nanoparticles that resemble native HDL in size and function. Classically, it is a two-component system including phospholipids and alpha helical proteins (mostly ApoA-I), but a number of molecular variations can be introduced in designing such particles. Generally, HDL mimetics prepared from full-length ApoA-I are termed as reconstituted HDL (rHDL, e.g., CSL112), and the ones using ApoA-I mimetics are called synthetic HDL (sHDL, e.g., ETC-642). The purpose of developing HDL mimicking nanoparticles in the context of vascular disease prevention is primarily (i) to restore the ‘functional’ HDL particle numbers, which decline in many disease settings and (ii) to improve HDL functionality. Such HDL infusion therapies enable, for example, the targeting of lipid-laden foam cells to restore optimal cholesterol efflux.

The intrinsic characteristics of native HDL have inspired therapeutic strategies for synthetic HDL nanoparticles. Their size, natural and specific targeting properties, long circulating time and stability in vivo make them an ideal drug delivery platform with endogenous protective effects. Apart from these significant advantages, one of the most beneficial aspects associated with synthetic HDL nanoparticles is the elimination of extensive biocompatibility validation. This reduces the tolerability issues which are associated with other classes of non-lipoprotein nanoparticles, their clearance being the most concerning question [36].

A majority of HDL nanodisc formulations rely on the amphipathic properties of ApoA-I and its interaction with lipids. Apart from incorporating full-length ApoA-I, HDL mimetics also include ApoA-I mimetic peptides, retaining the alpha helical structure that enables optimum nanodiscs configuration and cholesterol efflux [174]. Building upon the knowledge of HDL structure and function, HDL-inspired ApoA-I-based nanopharmaceutical drug development started off with the administration of full-length ApoA-I and progressed into lipidated versions of truncated ApoA-I mimetics, incorporating additional ligands for enhanced specificity. The lipid components of such exogenous HDL can be biocompatible synthetic phospholipids or lipids derived from the host when ApoA-I/ApoA-I mimetics mobilize phospholipids and/or cholesterol. Lipidation imparts the HDL nanoparticles a stable configuration and prolongs the circulation time. In principle, HDL mimetics are either starting material for pre-beta HDL formation in vivo (protein infusion only) or discoidal pre-beta HDL such as nanoparticles assembled before infusion (lipidated nanodiscs). The protein and lipid components currently being used for HDL mimetic nanodisc preparation are summarized in Table 1. The size and morphology of the HDL-mimicking nanoparticles depend on the stoichiometry of the interacting components, i.e., ApoA-I: lipid ratios. Investigating the interaction of ApoA-I with lipids has been the primary focus for generating optimally active and stable nanoparticles. Molecular dynamic simulations of HDL mimetic nanodiscs shed light on the assembly and stability of these particles [175]. Such atomistic approaches enable the better design, characterization and prediction of efficiency for mimetic peptide synthesis. Current strategies for HDL nanoparticle manufacture and assembly have been reviewed elsewhere [176].

### 6.1. Protein Components of HDL Mimetic Nanoparticles

#### 6.1.1. ApoA-I Full-Length Protein

ApoA-I, the most abundant protein in HDL with strong anti-inflammatory and antioxidant functions, naturally forms the basis for the synthesis of HDL nanoparticles [177]. ApoA-I interacts with endothelial and immune cells to exert a plethora of anti-inflammatory and protective homeostatic functions that are made use of by synthetic HDL infusion therapies. The initiation of HDL biogenesis is entirely dependent on the synthesis and release of ApoA-I and its cholesterol efflux capacity as discussed above. This 28.3 kD protein contains at least 10 alpha helices, a majority of them being class A amphipathic helices [178]. The secondary structure of ApoA-I is composed of a hydrophobic face that interacts with phospholipid acyl chains and a hydrophilic polar face that ensures water solubility. The negatively charged amino acids are in the center of the polar face, while the positively charged amino acids lie between the two faces. The positively charged amino acids such as arginine and lysine interact with the negatively charged polar heads of the phospholipids. The polar and negatively charged amino acid groups such as aspartic acid, glutamic acid and serine ensure stability in aqueous solvents [174].

ApoA-I used in HDL nanoparticle preparations is either isolated from human plasma or recombinant purified protein from *E. coli* or Chinese hamster ovary cells [36]. Recombinant proteins have the advantage that they can be scaled up, but they have to be put through rigorous purification processes, which ultimately makes the full-length protein use an expensive venture.

The measurement of cholesterol efflux capacity is considered as a standard for evaluating the efficiency of the HDL mimetic nanodiscs in vitro and in vivo. In circulation, the efflux of cholesterol due to the interaction of ApoA-I with ABCA-1 forms discoidal bilayer of HDL, usually containing two ApoA-I molecules and as many as five molecules per HDL based on the maturation phase of the particle [175]. The presence of an alpha helical secondary structure also allows for the exchange of ApoA-I between HDL subclasses which enables administered HDL mimetics to interact with the native HDL pool. This remodeling is also used as a readout for stability and activity ex vivo.

#### 6.1.2. ApoA-I Mimetic Peptides

Although the cholesterol efflux capacity of ApoA-I and ApoA-I mimetic peptides is similar, peptide derivatives have several advantages over the full-length protein. ApoA-I with a chain length of 243 amino acids is of limited use as a therapeutic agent. However, peptides with a length of only 18–37 amino acids have been synthesized based on the ApoA-I structure, which have been shown to possess some of the characteristics of full-length ApoA-I (Table 1). It is easier to obtain larger quantities of short truncated peptides by chemical synthesis platforms, thereby minimizing the time and cost of production. This also ensures a greater degree of purity and absence of host proteins or lipids that can lead to off-target effects and immunogenic reactions. The degree of amphipathicity in mimetic peptides correlates with the affinity for phospholipid interactions, and an increase in hydrophobicity increases the lipid affinity [174]. As much as this property enables design enhancement, this opens up several questions pertaining to biological activity of the synthesized peptides.

Most mimetic peptides are amphipathic helices with no or little primary homology to the full length ApoA-I. The 18A peptide was one of the first mimetic peptides designed and evaluated for confirming the amphipathic helix hypothesis (Table 1). Originally designed to characterize and study the behavior of amphipathic helices in nanodisc assembly [179,180], the 18A peptide also demonstrated anti-inflammatory properties [181,182]. For example, 18A was able to rapidly assemble with phospholipids into nanodiscs, solubilizing unilamellar vesicles and confirming the amphipathic helix hypothesis. They were shown to efflux cholesterol from macrophages and prevent monocyte adhesion to primary human endothelial cells. N and C terminal modifications with acetyl and amide groups were shown to enhance the lipophilicity and helicity of the peptide [181]. It was later renamed as 2F due to two phenyl alanine residues. Later on, 18A dimers were also tested for their stability and tunability for phospholipid interactions, demonstrating improved ‘features’ in bihelical peptides [183]. These initial experiments and pre-clinical findings confirm the stability, tunability and anti-inflammatory potential of mimetic peptides. 

Harnessing the scope of modifications in ApoA-I mimetics, the optimization of peptides continued with the development of 4F, 22A and 5A peptides with differences in the number, sequence and properties of amino acids used (Table 1). The 4F peptide has four phenylalanine residues, thereby increasing the hydrophobicity. Due to its stability in aqueous solutions, 4F is the most extensively studied ApoA-I mimetic peptide [184]. For example, 4F has been synthesized with both D and L amino acids to optimize the circulation time and anti-atherogenic properties. D-4F offers better resistance to proteolysis but shows lower oral bioavailability than the L-4F peptide [176,184,185]. The presence of alanine residues has been hypothesized to enhance ABCA1 specificity in mimetic peptides. For example, 5A, an asymmetrical bihelical peptide, with five alanine residues in its second helix and hydrophobic amino acids in the hinge region, has demonstrated increased cholesterol efflux capacity and tolerance due to lower toxicity [186,187]. Beyond ABCA-1-mediated cholesterol efflux, attempts have been made to develop mimetics that activate LCAT and increase cholesterol mobilization by engineering 22A peptides with ETC-642 [188]. Although ETC-642 does not have the advantages of 5A, it has shown tremendous preclinical and clinical potential in rabbit and mouse models, as discussed below.

Some of the recent mimetic peptides include Fukuoka University ApoA-I-mimetic peptide (FAMP), which is a 24 amino acid-long mimetic peptide. This single amphipathic helix and its C terminal modified form have demonstrated considerable cholesterol efflux ability and reduced plaque formation in ApoE^-/-^ mice [189,190]. Another mimetic peptide that has been based on a dimeric version of 4F, with a total of eight phenylalanine residues in the sequence of two 4Fs linked with a proline, has been studied in atherosclerosis and adipose tissue browning, demonstrating protective effects in in both ApoE^-/-^ atherosclerotic and db/db diabetic models [191]. This study also establishes the multiple protective roles of ApoA-I mimetics, underscoring the expanse of diseases that can be targeted with HDL nanotherapeutics. 

#### 6.1.3. Non-ApoA-I Mimetic Peptides

Apart from ApoA-I, few other apolipoproteins have also been developed in the form of mimetic peptides for the treatment of several inflammatory and metabolic disorders. 

The anti-inflammatory potential of ApoE mimetics has been evaluated, primarily in neurological and cardiovascular disease settings. Due to the presence of amphipathic helices in the C terminal region of ApoE, they also participate efficiently in lipid homeostasis by enhancing cholesterol mobilization. A modified version of 18A, once tested for N and C terminal modification, was also based partially on the ApoE structure. The receptor-binding structure of ApoE, combined with the 18A peptide, was used to create the dual-domain AEM28 peptide, which showed an increased removal of ApoB-containing lipoproteins. The AEM series of peptides have reached a phase IIa clinical trial for statin-intolerant patients [185,192]. ApoE mimetic peptide administration has demonstrated a dramatic reduction in plasma cholesterol in several dyslipidemic murine models [193,194]. One of the ApoE mimetics, CN-105, has shown neuroprotective potential and is currently being investigated in a clinical trial for the treatment of perioperative neurocognitive disorders. Currently, in phase II, preliminary results from phase I show a valid safety and efficacy profile [195].

ApoC-II also contains class A amphipathic helices in its N-terminal region, which is essential for lipoprotein binding and triglyceride (TG) clearance. New genetic studies have shown that it is causally linked to cardiovascular disease and thus is an active area for drug development [196]. Recently, a strategy to lower triglycerides with an ApoC-II mimetic peptide called D6PV, which activates lipoprotein lipase, the major TG-hydrolyzing enzyme in plasma, and antagonizes the TG-increasing effect of ApoC-III, was published [197]. Few pre-clinical studies that show LPL activation and antagonism to Apo C-III have been reviewed [89,185], but no clinical trials at present involve ApoC-II mimetic peptides.

### 6.2. Lipid Components of HDL Mimetic Nanoparticles

Manufactured HDL mimetics must meet certain physicochemical and biological criteria in order to be used for drug delivery and therapy. The nanoparticles must have similar dimensions, density and surface properties to HDL in order to be stable in the bloodstream over time and have non-toxic, non-immunogenic and biocompatible characteristics [198]. The synthesis process most commonly uses commercially available layer-forming phospholipids, but the role of the phospholipids is not limited to nanodisc stabilization. The phospholipids also act as substrates for LCAT. Some of the most common synthetic phospholipids include dipalmitoyl phosphatidylcholine (DPPC), 1,2-dimyristoyl-sn-glycero-3-phosphocholine (DMPC), palmitoyl-2-oleoyl-sn-glycero-3-phosphocholine (POPC), sphingomyelin (SM), 1-palmitoyl-2-linoleoyl phosphatidylcholine (PLPC), etc. The differences in these phospholipids exists in the degree of saturation, transition temperatures and other biochemical properties. An increasing degree of unsaturation in phospholipids reduces the efficiency of cholesterol efflux capacity of the HDL mimetics but increases the hydrolysis by LCAT.

However, optimization of the types and combinations of phospholipids for nanodisc assembly and functionality is not very advanced. One study compared the efficiency of a 5A peptide-based HDL nanodisc prepared using either POPC or SM [199]. In their in vitro and in vivo studies, the 5A-SM nanodisc was reported to be more anti-inflammatory and significantly more anti-atherogenic as compared to 5A-POPC. Moreover, they demonstrate significant differences in their ability to remodel murine HDL ex vivo. Interestingly, SR-B1-mediated cholesterol efflux was higher for 5A-SM, but ABCA-1-mediated cholesterol efflux was not significantly different. Lipid-22A interactions were stronger with SM than with POPC, which resulted in the formation of more homogeneous sHDL nanoparticles with longer in vivo circulation time [200].

In addition to pharmacokinetic properties, the composition of phospholipids also affects the pharmacodynamics characteristics of synthetic HDL nanodiscs. A recent study compared cholesterol mobilization, esterification, LCAT activity, HDL remodeling and peptide stability with different lipids in a 22A nanodisc [201]. Keeping the 22A peptide constant, DDPC, DMPC, DPC and POPC were incorporated to test the effect of transition temperatures and saturation of phospholipids in cholesterol efflux and the circulating time of the nanodisc. sHDL with variable phospholipid compositions showing significant differences in phospholipid pharmacokinetic properties, with DSPC sHDL demonstrating the longest half-life of 6.0 h relative to 1.0 h for POPC-based sHDL. This increase in half-life corresponded to a ~6.5-fold increase in the area under the curve for the mobilized cholesterol. The authors concluded that the phospholipid component in sHDL plays a major role in cholesterol mobilization in vivo and should not be overlooked in the design of future sHDL.

The incorporation of phosphatidylserine (PS) with POPC in HDL nanodiscs formulations has been shown to enhance the anti-inflammatory effects of the particles, independent of ABCA-1 and ABCG-1 interactions, but SR-B1-mediated anti-inflammatory effects were considerably enhanced [202]. The outcomes of these important studies [201,202] point to the need for disease-specific and stage-specific tailoring of sHDL phospholipid content for optimal therapeutic efficacy.

## 7. Anti-Inflammatory Effects of Synthetic/Reconstituted HDL Mimetic Nanoparticles

Most of the HDL infusion therapies remodel the native HDL pool, functionally augmenting the protective effects [203]. The primary readout for the efficacy of atheroprotective properties of HDL infusion therapies, apart from apart from vessel wall imaging for plaque volume reduction, has been the ability to mobilize cholesterol. Targeting monocytes and macrophages with HDL mimetics involves the mobilization of stored cholesterol in macrophages, modulation of recruitment of monocytes to the vessel wall, reduction in inflammatory cytokine expression, polarization and activation, which are discussed ahead. Table 2 summarizes the in vitro and in vivo results describing the atheroprotective properties of sHDL, rHDL and ApoA-I mimetic peptides, primarily targeting monocytes and macrophages.

### 7.1. Effects of HDL Nanodiscs Containing Full-Length ApoA-I on Monocytes/Macrophages

Apart from cholesterol efflux from foam cells, the administration of ApoA-I_M_ complexed with PC decreased the macrophage content of atherosclerotic plaques by 46% in ApoE-deficient mice, indicating that monocyte infiltration was efficiently suppressed [210]. The number of macrophages was also reduced after balloon injury in hypercholesterolemic rabbits treated with ApoA-I_M_ [211]. CER-001 suppressed inflammatory cytokine production upon LPS and TNF-α treatment in high cholesterol diet-fed LDL-receptor deficient mice. Of note, levels of monocyte chemoattractant protein-1 (MCP-1) were significantly decreased, suggesting that CER-001 interferes with monocyte recruitment by activated endothelial cells [212]. A single intravenous dose of CSL111 was shown to reduce the surface expression of CD11b on monocytes by 20% in a pre-clinical mouse model of myocardial infarction, reducing overall cardiac injury and inflammation [213]. In vitro treatments with CSL111 suppressed TLR2 expression and downstream nuclear factor kappa B (NF-κB) activation through the myeloid differentiation primary response 88 (MYD88) signaling arm in human macrophages [204]. CSL112 infusion decreased markers of plaque instability such as matrix metalloproteinase 9 and MCP-1 and pro-inflammatory cytokines such as interleukin-1β (IL-1β) [222].

The incorporation of PS in ApoA-I nanodiscs improved the anti-inflammatory effects in TNF-α and LPS-stimulated THP-1 monocytes. PS containing rHDL enhanced the anti-inflammatory activity by modulating Akt1/2/3- and p38 MAPK-mediated signaling through SR-B1 in stimulated macrophages. These data identified PS as a potent anti-inflammatory component capable of enhancing therapeutic potential of rHDL-based therapies. The efflux of cholesterol and Akt1/2/3- and p38 MAPK-mediated signaling through SR-BI were identified as the mechanism for the reduction in pro-inflammatory cytokine production and reduction in membrane lipid rafts abundance [202]. 

### 7.2. Effects of HDL Nanodiscs Containing ApoA-I Mimetic Peptides on Monocytes/Macrophages

ETC-642 was shown to reduce TNF-α-induced THP-1 monocyte adhesion and mRNA levels of VCAM-1, fractalkine, MCP-1 and the p65 subunit of NF-κB [188]. ETC-642 treatment significantly suppressed the targeting of TLR4 to lipid rafts in RAW264.7 macrophages [205].

The ApoA-I mimetic 4F peptide reduced macrophage content in lipid-rich lesions of dyslipidemic mice [219] and inhibited macrophage migration into the vessel wall due to decreased CD49d, CD11b, and CD11c expression [206]. The study also demonstrated that the 4F peptide reduced the phagocytic activity of human monocyte-derived macrophages, which is possibly due to lipid raft disruption. In the same study, it was shown that the 4F peptide promoted the M2 phenotype in macrophages, as confirmed by cytokine profiling. Moreover, 4F impaired the phosphorylation and nuclear translocation of NF-κBp65, CD14 expression, TLR-4 assembly and recycling, ameliorating LPS induced inflammation in human monocyte-derived macrophages [207]. In addition, 4F peptide treatment further reduced monocyte migration across cultured human aortic endothelial cells in ex vivo experiments, in pulmonary hypertension patients [208]. 

Apart from removing cellular cholesterol, mobilizing extracellular cholesterol is an important strategy for lesion size reduction and resolution. Interestingly, extracellular cholesterol microdomains were only mobilized by the 5A peptide complexed with SM, but not without lipidation, even in the absence of ABCA-1 [223]. Monocyte infiltration, levels of inflammatory monocytes in circulation and MCP-1, TNF-α and IL-6 levels in plasma were attenuated in a murine colitis model upon 5A peptide treatment [221].

The mimetic peptide Ac-hE18A-NH_2_ significantly attenuated IL-6 and MCP-1 secretion from LPS-stimulated THP-1 macrophages and promoted cholesterol efflux in a concentration-dependent manner [182].

Figure 3 summarizes the anti-inflammatory mechanism of HDL mimetics in monocytes and macrophages.

### 7.3. Anti-Inflammatory Effects of HDL Mimetics on Other Cell Types

The endothelial protective activities of native HDL are well known [224]. Endothelial cell activation is also profoundly modulated by HDL mimetics. Of note, VCAM-1 and ICAM-1 expression have been shown to decrease with synthetic HDL treatments, which is important in suppressing immune cell recruitment to the vessel wall [225,226]. The inability to mobilize endothelial progenitor cells from bone marrow for repair is associated with worse clinical outcomes in CAD [227]. A sub-population of 33 patients from the CSL111 ERASE trial with recent acute coronary syndrome was studied for reparative cardiovascular effects of HDL infusion. CSL111 infusion increased the number of circulating endothelial progenitor cells (CD34+ cells) [228]. Moreover, treatment with the reverse mimetic peptide D-4F led to an improvement in the endothelial progenitor cell numbers and function in a murine acute lung injury model via the PI3K/AKT/eNOS activation pathway [229]. In addition, 4F peptide treatment promoted the re-reendothelialization after oxidative injury in an SR-B1 dependent manner. It was shown that the 4F peptide increased eNOS activity, Akt phosphorylation, and SR-B1 expression in cultured human endothelial cells, protecting them from oxidative stress [230,231]. 

Plaque stability in atherosclerosis is highly dependent on platelet activation and aggregation at the lesion site. A clinical trial of CSL111 in a type 2 diabetes cohort demonstrated profound inhibitory effects on platelet activation, aggregation, and thrombus formation. This was dependent on ABCA-1 and was apparently caused by lipid raft collapse due to CSL111-induced cholesterol efflux [232]. However, when administered with dual antiplatelet agents, there were no significant additive effects of CSL112 on platelet function in a recent phase II trial. Therefore, it appears that there is no increased risk of bleeding after CSL112 administration [233].

## 8. HDL Nanotherapeutics: Existing Strategies and Clinical Trials

With a better understanding of the properties of synthetic and reconstituted HDL formulations and an increase in preclinical research results, several HDL mimetic nanoparticles and ApoA-I mimetic peptides have been tested in clinical trials.

### 8.1. ApoA-I Full-Length Protein: Reconstituted HDL (rHDL) Nanoparticles

#### 8.1.1. Clinical Trials with Non-Lipidated ApoA-I

The administration of ApoA-I isolated from pooled human plasma was attempted in humans owing to favorable data from mouse models of atherosclerosis by overexpressing ApoA-I [234,235,236,237,238]. In theory, this treatment mimics synthesis and release of nascent ApoA-I to some extent, which would mark the first step of HDL biogenesis. ApoA-I applied to the blood leads to the formation of pre-beta HDL through lipid incorporation via ABCA1, increasing the pre-beta HDL pool, and it later develops into fully mature HDL nanoparticles, analogous to native HDL biogenesis. The study confirmed its tolerability in humans, but functional protective readouts were inconclusive in terms of improved cholesterol efflux capacity, owing to high catabolism of the free protein [239]. This clearly highlights the need for the lipidation of ApoA-I prior to administration.

#### 8.1.2. Clinical Trials with Lipidated ApoA-I: rHDL Formulations

Another strategy of harnessing cardioprotective benefits of ApoA-I was inspired from the discovery of a mutant variant of ApoA-I in Milan, which was termed as ApoA-I Milano (ApoA-I_M_) thereafter. As an example of a single nucleotide polymorphism, the variant was characterized by the replacement of arginine by cysteine at position 173, which appears to be associated with better vascular health in carriers [240,241]. ApoA-I_M_ administration in atherosclerotic mice showed a significant decline in plaque volume and promoted the resolution of CAD [210,211]. Based on these promising effects, the first randomized controlled clinical trial was designed with the concept of administering full-length recombinant ApoA-I_M_ complexed with POPC, which was called ETC-216 [242]. The results were favorable in terms of reducing plaque volume and CAD regression. A follow-up trial suggested a strong correlation between decreased atheroma volume and compensatory vascular elastic remodeling in a specific, focal manner [243]. Furthermore, an increased understanding of the pharmacokinetic and pharmacodynamic attributes of such interventions were investigated [244]. A similar trial that used the ApoA-I_M_ variant for human trials was the MDCO 216 MILANO PILOT trial and its phase III trial. However, numerous side effects were reported in a dose-dependent manner in association with ETC-216 administration. The studies using POPC with the MDCO-216 formulation (ApoA-I_M_-POPC) reported a profound increase in ABCA-1 mediated cholesterol efflux in healthy and stable CAD patients [39,245,246,247]. The primary differences between the MDCO-216 and ETC-216 studies were the lipid composition of the formulations, protein to phospholipids ratio and the number of patients recruited (Table 1). Later, it was shown that the MDCO-216 formulation was better tolerated by the immune system than the ETC-216 formulation [248]. However, there is no drug based on the ApoA-I_M_ variant, as the trial has been interrupted several times, e.g., due to difficulties in producing large quantities of the full-length protein, increased production costs, change of company owners, etc. Commercially, the strategy of using recombinant full-length proteins has a few drawbacks. The synthesis of large quantities of the protein is expensive, and maintaining the quality can be a challenging aspect. A reduced half-life of about 3 h and rapid renal catabolism is another major challenge that needs to be overcome. The immuno-stimulation problems of ETC-216 were attributed to presence of host antigens, which puts stricter regulation criteria on the concept of full-length, ApoA-I based therapies.

Following up, wild-type ApoA-I was evaluated for its possible therapeutic potential in HDL-mimicking nanodisc formulations. Combining recombinant ApoA-I with two different types of phospholipids, sphingomyelin and dipalmitoyl phosphoglycerol, CER-001 was synthesized and tested in clinical trials funded by Cerenis Pharmaceuticals. CER-001 has shown considerable potential in mouse models of atherosclerosis [212,249]. In the first double-blind study of multiple intravenous doses of 3 mg/kg, 6 mg/kg, or 12 mg/kg of CER-001 in 507 randomized subjects, the results were not significant in terms of atheroma volume reduction and cumulative coronary stenosis score [250]. A follow-up trial of with serial infusions of CER-001 at a lower dose of 3 mg/kg was designed with 272 patients with acute coronary syndrome, again failing to show a significant improvement in atherosclerotic plaque regression [251]. In 2017, CER-001 was re-studied in a phase I dose optimization clinical trial. The study showed that a single infusion of 45 mg/kg is safe and efficiently mobilizes cholesterol in 32 healthy volunteers [252]. In patients with homozygous familial hypercholesterolemia, prolonged CER-001 infusions demonstrated beneficial effects by significantly reducing mean carotid vessel wall area, suggesting a possible reversal in arterial wall remodeling [253]. Furthermore, in a phase II trial in patients with familial hypoalphalipoproteinemia, CER-001 significantly increased cholesterol mobilization [254]. However, CER-001 did not improve carotid atherosclerosis in patients with genetically very low HDL levels [255]. Overall, CER-001 proved to be well tolerated, but the dose required to potentially see beneficial effects of CER-001 in CAD remains uncertain.

Studies with isolated human ApoA-I were continued by Commonwealth Serum Laboratories (CSL) Behring, testing ApoA-I conjugated with soy phosphatidylcholine in the ERASE trial. Their first formulation, CSL111, was evaluated as short-term infusions in 171 patients testing a lower and a higher dose (40 and 80 mg/kg, respectively). However, the results were not significant in atheroma characterization and coronary angiography results when compared to the placebo group [256]. Moreover, the treatments were associated with a significant increase in bilirubin in the higher dose group, and hypotension was seen in the lower dose group. Following up on the formulation, CSL112 was first tested in rabbits [257] and was later administered in single and multiple doses in healthy subjects, showing a dramatic and immediate increase in ApoA-I levels, cholesterol efflux capacity and cholesterol esterification [258]. In a randomized, multicenter, double-blind phase IIa study in patients with stable atherosclerosis, a single infusion of CSL112 in groups of three doses, 1.7 g, 3.4 g, or 6.8 g, was well tolerated, and the treatment resulted in significant cholesterol mobilization [37,214]. The first part of the AEGIS I study conducted in approximately 1200 CAD patients who had recently experienced an AMI event was designed to characterize the dosing and safety profile of CSL112 [37]. CSL112 at a dose of 6 g dramatically increased cholesterol efflux to about 300%, which was much more compared to all other previous studies [215]. The efflux capacity was reported to be similar in healthy and stable atherosclerotic patients [216] that remained unaffected in conditions of moderate underlying renal complications [217]. This outcome is of specific relevance, as kidney disease severely limits HDL functionality [95,259]. In the CSL112-related studies, significant emphasis has been placed on improving study design, gaining mechanistic insights, and continuously monitoring off-target/additive drug effects and co-treatment options [218,233,260]. The CSL112-based studies are one of the few that seek to determine and understand native HDL remodeling and pre-beta HDL formation with exogenous sHDL nanotherapy. The potential beneficial effects of CSL112 are being investigated in an ongoing multicenter, double-blind, randomized, placebo-controlled phase III study (AEGIS-II study) including about 17,400 participants [261]. The primary objective of AEGIS II is to investigate the efficacy of CSL112 in reducing the risk of death, myocardial infarction, or stroke from the time of randomization to 90 days in patients with acute coronary symptoms.

### 8.2. Clinical Trials with ApoA-I Mimetics

Despite the protective and anti-inflammatory effects of full-length ApoA-I, there are disadvantages associated with them, such as high production costs and quality control. The first mimetic peptide developed to study the ApoA-I nanodisc formation, 18A, gave deeper understanding of the HDL nanodisc assembly and paved the way for introducing nuances in the structural design to enhance vascular protective features [179,262]. 

The sHDL nanodisc formulation ETC-642 was developed by Esperion and has been extensively tested for its anti-atherogenic characteristics [188,263,264]. The ETC-642 complex consisted of ESP-2418, a 22 amino acid ApoA-I mimetic peptide complexed to two phospholipids, sphingomyelin and DPPC. The trials in human subjects proceeded to phase I with good tolerability and evidence of LCAT activation [264]. However, the study was discontinued in 2006 due to Pfizer ownership. Following this first mimetic peptide leading to a clinical trial [265], several studies reported anti-inflammatory properties of ETC-642 in mice and in vitro [205,264,266,267]. Currently, ETC-642 is not being tested in clinical trials.

The 4F peptide developed by Novartis was tested, for the first time, in oral formulations [268,269,270,271,272]. The 4F peptide was formulated with both D- and L-amino acids, which had similar biological characteristics but different bio-availabilities due to different helicity and amphipathicity. Here, D-4F was well tolerated in a phase I trial, and preliminary data suggested a potent anti-inflammatory activity [273,274]. D-4F showed promising results and the study indicated the intestine as the main site of action with plasma concentrations of D-4F being independent of the route of administration. However, L-4F did not show significant positive results in CAD prevention [275]. In vitro studies have demonstrated 4F-mediated HDL remodeling [276]. It should be emphasized here that the peptides were not lipidated before administration.

The ApoA-I mimetic peptide 5A produced by KlineMed. Inc. is currently another promising candidate for atherosclerosis prevention, as shown by in vitro and in vivo data from mouse and primate models [209,220,277]. 5A peptide incorporated into nanodiscs with SM has demonstrated high cholesterol efflux capacity, anti-inflammatory and anti-oxidant properties in the animal models and is under safety profile and pharmacokinetic evaluation in a phase I clinical trial.

## 9. Future Outlook and Perspectives

Given the abundance of reports favoring beneficial outcomes of HDL on monocytes and macrophages [278], several clinical trials attempted a successful translation of HDL infusion therapies. An overwhelming number of studies highlight the role of HDL in cardiovascular disease. Continuous optimization efforts are required to improve the pharmacological characteristics, such as the half-life and targeting accuracy of synthetic HDL nanoparticles, as demonstrated by recent reports on modified and loaded HDL nanodiscs. For example, the incorporation of polyethylene glycol (PEG) into synthetic HDL mimetics increased cholesterol mobilization in rats [279]. Another modification that has been studied in sHDL is the addition of polylactic-co-glycolic acid (PLGA). In addition to atheroprotective characteristics, the addition of PLGA to sHDL was shown to confer selective localization in atherosclerotic plaques [280]. PLGA, phospholipids, and mimetic 4F peptide in combination with triphenylphosphine in quantum dot nanoparticles also showed improved diagnostic potential in CAD due to the differential recognition of apoptotic and healthy cells [281]. Metal-core-based ApoA-I nanoparticles that mimic HDL functions are also a promising diagnostic tool for the detection of atherosclerotic plaques. Gold-based sHDL have been shown to enhance reverse cholesterol transport pathways and plaque analysis. ApoA-I-based gold sHDL stabilized cholesterol esters in the core, allowing sustained LCAT activation and the continuous efflux of cholesterol [282]. The use of gold and iron in sHDL also showed diagnostic advantages in computed tomography (CT) scans by improving the signal-to-noise ratio during detection [35]. Moreover, the addition of endothelial-specific ligands to the sHDL platform enhances the targetability of HDL nanotherapeutics. Various peptides with the VHPKQHR motif (VHPK peptides) have been widely used in developing inflamed endothelial targeting imaging agents and drug delivery systems. The endothelial VCAM-1-targeting VHPK peptide was conjugated to the surface of a 22A-nanodisc, demonstrating increased biodistribution, which is expected to increase the half-life of ApoA-I-based therapies [283].

Apart from atherosclerosis, HDL mimetics show promising outcomes in murine models of sepsis and acute respiratory distress syndrome [205,284]. Along with confirming the well-known LPS binding capacity of rHDL shown in human endotoxemia [285], these studies also demonstrate additional anti-inflammatory effects attributed to TLR4 targeting in the lipid raft and improving HDL-PON-1 levels. In a 1997 clinical study, rHDL was shown to modulate the procoagulant state induced by LPS [286]. In diabetes, the removal of cholesterol from the platelet membrane by rHDL was identified as one of the anti-thrombotic mechanisms, highlighting its effectiveness in reducing diabetes-associated thrombotic complications [232]. HDL treatment in a rat model of embolic occlusion showed beneficial outcomes by modulating neutrophil recruitment [287]. HDL-C levels are a prognostic marker in pulmonary embolism [288,289], suggesting the potential benefit of HDL-mimetic therapies. 

The effect of CER-001 as a salvage therapy was recently evaluated in four COVID-19 patients, three out of which showed a decline in cytokine storm and hence, recovery [290]. CER-001 administration also showed a significant reduction in HDL-SAA content, improving the native HDL–cargo characteristics [291]. A decline in C-reactive protein and inflammatory cytokines, especially during initial days of administration, was also shown in the same study.

In pulmonary hypertension, the so-called “inflammatory index” of HDL was evaluated ex vivo. HDL pre-treated monocyte migration across cultured human aortic endothelial cells was quantified with/without 4F peptide addition. The 4F peptide treatment reduced monocyte migration and improved HDL quality [208].

Vascular inflammation also contributes to the development of abdominal aortic aneurysm (AAA), and HDL composition and functions have been reported to be altered in the disease [292,293]. The use of modified HDL as a surrogate biomarker of oxidative stress in AAA has also been proposed [294]. The oxidation of ApoA-I and altered ApoA-I/HDL-C levels in AAA are associated with disease severity [295,296]. Treatment with freshly isolated HDL showed antithrombotic and protective effects in *P. gingivalis*-induced AAA in mice [297]. These positive outcomes appeared to be linked to a reduction in neutrophil activation and an improvement in the clearance of bacteria.

Collectively, these reports broaden the scope of therapeutic properties of HDL mimetics in several diseases. HDL nanoparticles and ApoA-I mimetics may help improve HDL quality in numerous inflammatory settings. However, the full pharmacological potential of HDL mimetics in other vascular diseases, severe inflammation and sepsis is yet to be thoroughly explored.

## 10. Conclusions

The focus of new HDL therapeutics has shifted from increasing circulating HDL-C concentrations to improving HDL function, particularly HDL cholesterol efflux capacity, which is impaired in patients with cardiovascular disease. In summary, there is now sufficient evidence from pre-clinical and clinical studies that the enhancement/restoration of cholesterol efflux by HDL mimetic nanoparticles is a valid strategy to stabilize atherosclerotic plaque through various anti-inflammatory/immunoregulatory processes. In addition, the immunomodulatory effects of HDL nanoparticles on monocytes and macrophages enable multiple cardioprotective properties that improve vascular health. Data from the ongoing phase III AEGIS-II trial testing CSL112 may represent a breakthrough in the treatment paradigm for acute myocardial infarction, particularly in the immediate weeks and months after the event, when patients are at high risk for the recurrence of ischemic events.

## Figures and Tables

**Figure 1 pharmaceutics-15-01504-f001:**
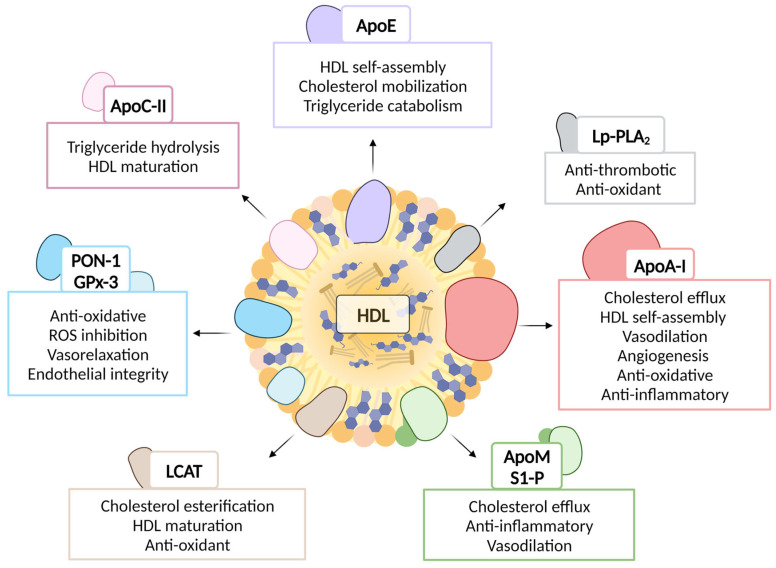
Protective features of native HDL nanodisc components in cardiovascular disease. Apolipoprotein (Apo), lipoprotein-associated phospholipase A_2_ (Lp-PLA_2_)*,* sphingosine-1-phosphate (S1P), lecithin-cholesterol acyltransferase (LCAT), paraoxonase-1 (PON-1), glutathione peroxidase-3 (GPx-3), reactive oxygen species (ROS).

**Figure 2 pharmaceutics-15-01504-f002:**
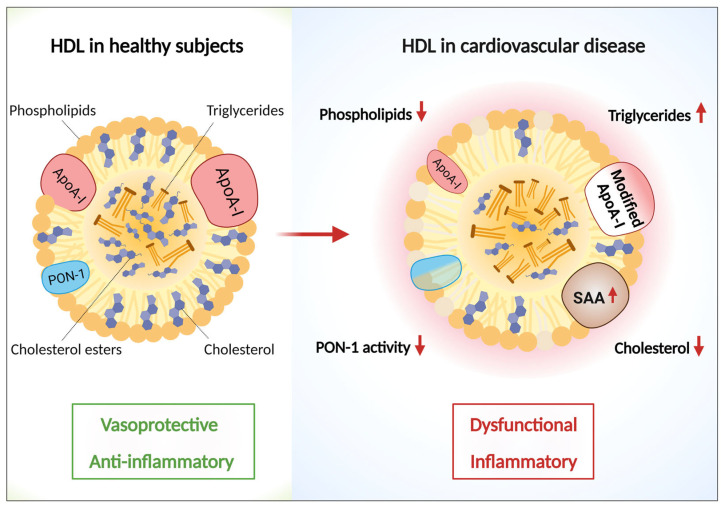
Modifications of selected HDL components in cardiovascular disease pathophysiology. Cardiovascular disease renders HDL dysfunctional, leading to structural modifications and functional losses. Apolipoprotein (Apo), HDL cholesterol (HDL-C), serum amyloid A (SAA), and paraoxonase-1 (PON-1).

**Figure 3 pharmaceutics-15-01504-f003:**
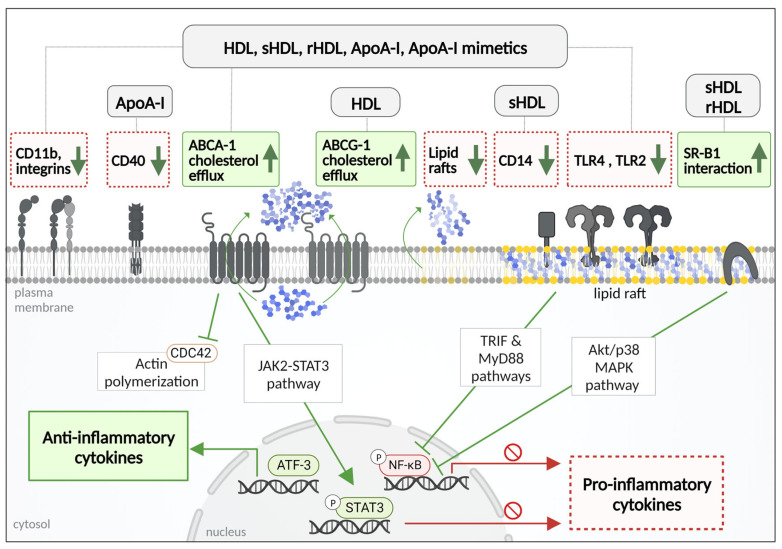
Anti-inflammatory mechanisms of native HDL nanoparticles, sHDL, rHDL and ApoA-I mimetic peptides in monocytes and macrophages. Nuclear factor kappa B (NF-κB) activation, myeloid differentiation primary response 88 (MYD88), Toll-like receptors (TLRs), TIR-domain-containing adapter-inducing interferon-β (TRIF), mitogen-activated protein kinase (MAPK), Janus kinase-signal transducer and activator of transcription (JAK-STAT), activating transcription factor 3 (ATF3), cell division control protein 42 (CDC42).

**Table 1 pharmaceutics-15-01504-t001:** Summary of sHDl, rHDL and ApoA-I mimetics in clinical trials.

HDL Mimetic	Protein Type	Peptide Sequence (P)	Phospholipid (L)	Nanoparticle Size	Ratio (P/L)	Administration	Patients Recruited	Clinical Trial	Major Findings	Reason to Discontinue Clinical Trial
ETC-216	Recombinant ApoA-I_M_	ApoA-I Milano	DPPC	7–30 nm	1:2.7 (*w*/*w*)	5 weekly, intravenous, 15 mg/kg and 45 mg/kg	57	Milano, Phase II	Significantly decreased atheroma volume	Change of company ownership; protein production and quality constrain
MDCO-206	Recombinant ApoA-I_M_	ApoA-I Milano	POPC	7–30 nm	1:1.1 (*w*/*w*)	5 weekly, intravenous, 20 mg/kg	126	MILANO-PILOT, Phase III, NCT02678923	80.4% increase in cholesterol efflux, no increment in plaque regression	Change of company ownership; protein production and quality constrain
CER-001	Recombinant ApoA-I	ApoA-I native	SM, DPPG	7–13 nm	1:2.7 (*w*/*w*)	10 weekly, intravenous, 3 mg/kg with statins and 6 weekly, intravenous, 12 mg/kg	507 293	CHI SQUARE, Phase II NCT01201837 and CARAT, Phase II NCT2484378	Failed to promote coronary atherosclerosis regression	Insufficient beneficial effects in CAD
CSL111	Plasma purified human ApoA-I	ApoA-I native	Soy PC	7–30 nm	1:150 molar ratio	4 weekly, intravenous, 40 mg/kg and 80 mg/kg	111	ERASE Phase II, NCT00225719	Significant improvement in the plaque characterization index, elevation in liver enzymes in 80 mg/kg group	Modified and improved to CSL112
CSL112	Plasma purified human ApoA-I	ApoA-I native	Soy PC	7–13 nm	1:55 molar ratio	4 weekly, intravenous, 2 g and 6 g	1258	AEGIS-I, Phase II, NCT02108262	No renal/hepatic toxicity, well tolerated	_
CSL112	Plasma purified human ApoA-I	ApoA-I native	Soy PC	7–13 nm	1:55 molar ratio	4 weekly intravenous, 6 g	17400	AEGIS-II, Phase III, NCT03473223	Results yet to be concluded	_
ETC-642	ApoA-I mimetic- 22 amino acids, single helix	P-V-L-D-L-F-R-E-L-L-NE-L-L-E-AL-K-Q-K-L-K	DPPC and SM	7–13 nm	1:1:1 molar ratio	4 weekly intravenous, up to 30 mg/kg	28	Phase I	Safe and well tolerated, LCAT activation	Change of company ownership
D-4F	ApoA-I mimetic peptide, 18 amino acids, bihelical	Ac-D-W-F-K-A-F-Y-D-KV-A-E-K-F-K-E-A-FNH 2	Non- lipidated	-	-	Three separate groups, sequential, ascending doses 100 mg, 300 mg, or 500 mg, oral	104	Phase I, NCT00907998	Increased hydrophobicity, blocks absorption of production of oxidized lipids, anti-inflammatory	Insufficient bioavailability
L-4F	ApoA-I mimetic peptide, 18 amino acids long, bihelical	4F enantiomer, L-amino acids	Non- lipidated	-	-	Single and 7-daily infusions, Intravenous, subcutaneous	176	Phase I, NCT00568594	Similar to 4F but shorter circulating time due to proteolysis	Insufficient HDL-function-biomarker improvements
5A	ApoA-I mimetic peptide, 37 amino acids, bihelical	D-W-L-K-A-F-Y-D-K-V-A-E-K-L-K-E-A-F-P-D-W-A-K-A-A-Y-D-K-A-A-E-K-A-K-E-A-A	SM	-	1:8 (primate study)	5 weekly, intravenous, 2.5, 5.0, 10.0 and 20.0 mg/kg	64, currently recruiting	Phase I, NCT04216342	ABCA-1 specificity, results yet to be concluded	_

Legends: dipalmitoyl phosphatidylcholine (DPPC), 1,2-dimyristoyl-sn-glycero-3-phosphocholine (DMPC), dipalmitoyl phosphatidylglycerol (DPPG), palmitoyl-2-oleoyl-sn-glycero-3-phosphocholine (POPC), 1-palmitoyl-2-linoleoyl phosphatidylcholine (PLPC), phosphatidylcholine (PC), sphingomyelin (SM).

**Table 2 pharmaceutics-15-01504-t002:** Summary of anti-inflammatory and atheroprotective properties of sHDL, rHDL and ApoA-I mimetic peptides in vitro and in vivo.

HDL Mimetic	In-Vitro	Primary Effects	References
CSL111	Human monocyte-derived macrophages	Cholesterol efflux, TLR2 suppression, NF-κB inactivation	[204]
ETC-642	THP-1 monocytes	Reduction in monocyte adhesion, MCP-1 and NF-κB activation	[188]
ETC-642	RAW264.7 macrophages	Reduction in TLR4 targeting to lipid rafts	[205]
PS-ApoA-I	THP-1 monocytes and human monocyte derived macrophages	Cholesterol efflux, reduction in inflammatory cytokine production and lipid raft abundance	[202]
4F peptide	Human monocyte-derived macrophages, THP-1 monocytes	Cholesterol efflux, M2 polarization, impaired phosphorylation and nuclear translocation of NF-κBp65, phagocytosis, inflammatory surface markers, CD14 expression, TLR4 assembly and recycling, reduced monocyte adhesion and migration	[206,207]
4F peptide	Human monocytes	Reduced chemotaxis	[208]
18 A peptide	THP-1 macrophages, human umbilical vein endothelial cells	Cholesterol efflux, M2 polarization, reduction in reduction in inflammatory cytokine production and MCP-1 secretion	[182]
5A-POPC/5A-SM-DPCC	RAW 264.7 macrophage	Cholesterol efflux	[209]
**HDL Mimetic**	**in-vivo**	**Primary Effects**	**References**
ApoA-I_M_-PC	ApoE-deficient mice	Cholesterol efflux, reduced macrophage content in plaques	[210]
ApoA-I_M_	Hypercholesteremic rabbits	Cholesterol efflux, reduced intimal thickening and macrophage counts	[211]
CER-001	LDL receptor deficient mice	Cholesterol efflux, reduction in MCP-1 and inflammatory cytokine production	[212]
CSL-111	Murine model of myocardial infarction	Cholesterol efflux, reduced CD11b on circulating monocytes	[213]
CSL-112	Clinical trials	Cholesterol efflux, reduced plaque instability markers and anti-inflammatory cytokines, increased cholesterol esterification by LCAT	[204,214,215,216,217,218]
4F peptide	Dyslipidemic mice model of CAD	Cholesterol efflux, reduction in macrophage content in lipid-rich lesions	[206,219]
5A-PLPC	Collared arteries in rabbit	Cholesterol efflux, reduced oxidative stress	[220]
5A-POPC/ 5A-SM-DPCC	ApoE-deficient mice	Cholesterol efflux, reduction in plaque coverage	[209]
5A peptide	Murine colitis model	Reduced monocyte infiltration	[221]

## Data Availability

Not applicable.

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
