# Peer review of "A Current Update on the Role of HDL-Based Nanomedicine in Targeting Macrophages in Cardiovascular Disease"

_pharmaceutics, 2023, doi:10.3390/pharmaceutics15051504_

Round 1
Reviewer 1 Report
This is an excellent and comprehensive review on HDL structure, function and therapeutic potential of synthetic HDL nanoparticles. Corrections of several parts of the manuscript will improve it:
1) References 70 and 95 appear to be the same. Maybe different chapters are to be referenced.
2) Figure 2: Light green, green, light blue, purple and light purple shapes are not explained.
3) A Table is necessary, summarizing the results of the effect of apoA-I mimetics and synthetic HDL particles on HDL atheroprotective functions, as assessed by cell based and in vitro assays.
4) Another table summarizing the results of the effect of apoA-I mimetics and synthetic HDL particles on animal models would be also extremely useful for the reader.
5) In Table 1 another column should be added stating the reasons for the discontinuation of the various clinical trials.
6) References should be checked for formatting. Several of them are formatted differently from the majority of references. Furthermore, some of them are incomplete. Characteristic examples of references that need correction are: 1, 2, 10, 30, 34, 35, 60, 62, 68, 74, 109, 255.
Author Response
Review Report 1:
We would like to thank the reviewer for their expertise and excellent comments. We hope that the updated manuscript addresses all the comments and incorporates all the suggestions as expected.
1) References 70 and 95 appear to be the same. Maybe different chapters are to be referenced.
References were duplicated, and have been corrected.
2) Figure 2: Light green, green, light blue, purple, and light purple shapes are not explained.
According to the reviewer’s suggestion, we have altered the figure design to eliminate the need for further labeling.
3) A Table is necessary, summarizing the results of the effect of apoA-I mimetics and synthetic HDL particles on HDL atheroprotective functions, as assessed by cell-based and in vitro assays. 4) Another table summarizing the results of the effect of apoA-I mimetics and synthetic HDL particles on animal models would be also extremely useful for the reader.
According to the reviewer’s suggestion, Table 2 is now added to the manuscript, addressing points 3) and 4) together, describing both in-vitro and in-vivo results mentioned in the manuscript.
5) In Table 1 another column should be added stating the reasons for the discontinuation of the various clinical trials.
We have added another column in Table 1 stating the reasons for the discontinuation of respective clinical trials.
6) References should be checked for formatting. Several of them are formatted differently from the majority of references. Furthermore, some of them are incomplete. Characteristic examples of references that need correction are 1, 2, 10, 30, 34, 35, 60, 62, 68, 74, 109, 255.
All the references have been checked and reformatted.
Reviewer 2 Report
The interesting proposed review performs a deep and comprehensive update of current terapies based on HDL components or properties to target macrophages in CVD. The manuscript is well structured with detailed table and figures. The references list is well documented and a little bit large, but the topic on HDL-based therapeutic approaches is widely studied.
One minor suggestion is to improve the quality and resolution of the figures; also, the size of text in all figures could be increased for an easier reading.
Secondly, the table need to be edited for an easier reading, restructured and repositioned in the text, maybe in a landscape form, if accepted, accordingly to the journal requirement.
Author Response
We would like to thank the reviewer for his/her comments. We hope that the updated manuscript addresses all the comments and incorporates all the suggestions as expected.
- One minor suggestion is to improve the quality and resolution of the figures; also, the size of the text in all figures could be increased for easier reading.
According to the reviewer’s suggestion, we have increased the size of text and resolution of the figures. Initially, it was imported in 300 dpi, which is now 600 dpi. Additionally, we have modified the color scheme of Figure 3, which helps improve visualization and resolution.
- Secondly, the table needs to be edited for easier reading, restructured, and repositioned in the text, maybe in a landscape form, if accepted, accordingly to the journal requirement.
According to the reviewer’s suggestion, the table has been formatted for better visualization.
Reviewer 3 Report
In this review the authors summarized the evidence on the effects of plasma-derived and synthetic HDL on monocytes and macrophages in the context of atherosclerotic cardiovascular disease.
Here are some comments to improve manuscript clarity.
1. At lines 33-36, the authors should also indicate the inverse relationship between HDL-C and CVD risk. Otherwise, the sentence could be difficult to understand for a reader not expert in the field.
2. At lines 103-105, the authors should consider supporting the presence and prevalence of distinct macrophage subclasses (and of other immune cells in general) also with data from plaque scRNA-seq. This distribution into distinct subclasses is not taken into account in the following sections of the manuscript.
3. In figure 1, the authors should consider including the antioxidant role also for LCAT and Lp-PLA2. In addition, the ability of apoA-I and S1P in HDL to mediate vasodilation should be cited as well.
4. HDL is used as singular or as plural term in different sections of the manuscript.
5. Section 2.1.1 on apolipoproteins: the authors should clearly indicate whether data are on unlipidated or HDL-associated apoA-I. Please check sentence at lines 185-187 for clarity. The authors should also consider highlighting the role of apoE, CII and CIII in HDL and not only their activity on lipid metabolism in general.
6. Lines 210-216: the authors should consider mentioning how PON inclusion into HDL affects its stability/activity.
7. Line 229: the authors should clarify the meaning of “between lipoprotein and chylomicron subclasses”.
8. Lines 231-232: the authors should clarify whether these data are from humans and/or from animal models.
9. Check sentence at lines 301-303 for clarity.
10. Check “They and others” at line 311.
11. At line 311, “HDL therapy” is indicated. However, the concept of HDL therapy has not been introduced yet. The authors should consider to better explain the experimental approach used in the cited papers in order to clarify what HDL therapy means.
12. Table 1 should have a legend with the abbreviations used.
13. Line 481: would 18-37 aminoacids be a better range (5A)?
14. Line 565 and others: please consistently use DPPC to abbreviate dipalmitoyl phosphatidylcholine.
15. Lines 722-726: the authors should check for repeated sentences.
16. Carefully check the reference list.
Minor rephrasing is suggested to improve clarity.
Author Response
We would like to thank the reviewer for his/her helpful and excellent comments. We hope that the updated manuscript addresses all the comments and incorporates all the suggestions as expected.
1) In lines 33-36, the authors should also indicate the inverse relationship between HDL-C and CVD risk. Otherwise, the sentence could be difficult to understand for a reader not expert in the field.
According to the reviewer’s suggestion, we have added one line to improve the message, lines: 38-39
2) At lines 103-105, the authors should consider supporting the presence and prevalence of distinct macrophage subclasses (and of other immune cells in general) also with data from plaque scRNA-seq. This distribution into distinct subclasses is not taken into account in the following sections of the manuscript.
According to the reviewer’s suggestion, we have mentioned input from RNA seq studies, lines: 114-120.
3) In Figure 1, the authors should consider including the antioxidant role also for LCAT and Lp-PLA2. In addition, the ability of apoA-I and S1P in HDL to mediate vasodilation should be cited as well.
We thank the reviewer for this important comment. We have upgraded Figure 1 with additional information.
4) HDL is used as a singular or as a plural term in different sections of the manuscript.
The singular form is now consistently used.
5) Section 2.1.1 on apolipoproteins: the authors should clearly indicate whether data are on unlipidated or HDL-associated apoA-I. Please check the sentence at lines 185-187 for clarity. The authors should also consider highlighting the role of apoE, CII, and CIII in HDL and not only their activity on lipid metabolism in general.
We thank the reviewer for this important comment. According to the reviewer’s suggestion, we have indicated HDL-association/lipidation status of ApoA-I, lines: 183-186 and paraphrased the suggested lines. In the sections that explain mechanisms and clinical trials, the sub-section heading clarifies the lipidation status of ApoA-I/mimetic peptide.
We have expanded the apolipoprotein section to highlight the role of ApoE, CII, and CIII on HDL, lines: 190-238
6) Lines 210-216: the authors should consider mentioning how PON inclusion into HDL affects its stability/activity.
We thank the reviewer for this comment. We have expanded our text in this regard, lines: 249-263
7) Line 229: the authors should clarify the meaning of “between lipoprotein and chylomicron subclasses”.
We thank the reviewer for this comment We made necessary changes, line 276.
8) Lines 231-232: the authors should clarify whether these data are from humans and/or from animal models.
According to the reviewer’s suggestion, we have modified the sentence, line 279.
9) Check the sentence at lines 301-303 for clarity.
Paraphrased, lines: 350-352.
10) Check “They and others” at line 311.
Paraphrased, line: 360
11) At line 311, “HDL therapy” is indicated. However, the concept of HDL therapy has not been introduced yet. The authors should consider better explaining the experimental approach used in the cited papers in order to clarify what HDL therapy means.
According to the reviewer’s suggestion, we have removed the word ‘therapy’ as the treatment guidelines using HDL are still not available, awaiting the clinical trial results. The sentence has been modified to improve clarity, lines: 360-362.
12) Table 1 should have a legend with the abbreviations used.
We have included a legend in Table 1-footer.
13) Line 481: would 18-37 amino acids be a better range (5A)?
Changed to 18-37, line: 543
14) Line 565 and others: please consistently use DPPC to abbreviate dipalmitoyl phosphatidylcholine.
Necessary changes made, DPPC abbreviation consistently used.
15) Lines 722-726: the authors should check for repeated sentences.
We thank the reviewer for this comment. Repeated sentences have been removed.
16) Carefully check the reference list.
All the references have been checked and reformatted.
17) Comments on the Quality of English Language-Minor rephrasing is suggested to improve clarity.
We have paraphrased some sentences, as mentioned above.
Reviewer 4 Report
The review "A current update on the role of HDL-based nanomedicine in targeting macrophages in cardiovascular disease" represents an interesting literature summary regarding HDL nanotherapeutics in vascular disorders therapy. The present paper included relevant references and facts, being adequate for publishing. Just two suggestions for the authors:
- Please double-check abbreviations (first appearance)
- In section 8. Future Outlook and Perspectives, the authors pointed out the pharmacological potential of HDL mimetics in sepsis and other vascular diseases besides atherosclerosis. I suggest that this section should be extended with more animal (and perhaps clinical studies) studies regarding various vascular disorders (perhaps aneurysm, thrombosis, embolism, etc.)
Author Response
We would like to thank the reviewer for his/her expertise and comments. We hope that the updated manuscript addresses all the comments and incorporates all the suggestions as expected.
- Please double-check abbreviations (first appearance)
According to the reviewer’s suggestion, we have checked the abbreviations for the first appearance (for e.g. Line 421) We also added a legend to Table 1 to improve upon that issue.
- In section 8. Future Outlook and Perspectives, the authors pointed out the pharmacological potential of HDL mimetics in sepsis and other vascular diseases besides atherosclerosis. I suggest that this section should be extended with more animal (and perhaps clinical studies) studies regarding various vascular disorders (perhaps aneurysm, thrombosis, embolism, etc.)
According to the reviewer’s suggestion, we have extended the Future Outlook and Perspective section with necessary information and references, lines: 925-954.
Round 2
Reviewer 1 Report
The updated manuscript addresses all my comments and incorporates my suggestions.